

# Identification of potential insect ecological interactions using a metabarcoding approach

Nicole D. Borsato[1,*], Katherine Lunn[1,*], Nina R. Garrett[1],
Alejandro José Biganzoli-Rangel[1], Daniel Marquina[1,2], Dirk Steinke[3],
Robin Floyd[3] and Elizabeth L. Clare[1]

[1] Biology, York University, Toronto, ON, Canada
[2] AllGenetics, Perillo, Spain
[3] Centre for Biodiversity Genomics, University of Guelph, Guelph, Ontario, Canada
* These authors contributed equally to this work.

## ABSTRACT

Species interactions are challenging to quantify, particularly when they happen cryptically. Molecular methods have become a key tool to uncover these interactions when they leave behind a DNA trace from the interacting organism (*e.g.*, pollen on a bee) or when the taxa are still present but morphologically challenging to identify (*e.g.*, microbial or fungal interactions). The decreasing costs of sequencing makes the mass analysis of thousands of target species possible. However, the challenge has shifted to selecting molecular markers which maximize information recovery while analyzing these data at broad biological scales. In this manuscript we use model arthropod groups to compare molecular markers and their analysis across life stages. We develop protocols for two ecologically and economically devastating pests, the spongy moth (*Lymantria dispar dispar*) and the emerald ash borer (*Agrilus planipennis*), and a group of pollinators including bees and wasps which regularly deposit eggs in "bee hotels" where the larvae develop. Using Illumina MiSeq and Oxford Nanopore MinION platforms we evaluate seven primer pairs for five molecular markers which target plants, fungi, microbes, insects, and parasitic phyla (*e.g.*, nematodes). Our data reveals hundreds of potential ecological interactions and establishes generalized methods which can be applied across arthropod host taxa with recommendations on the appropriate markers in different systems. However, we also discuss the challenge of differentiating co-occurring DNA signals and true ecological interactions, a problem only starting to be recognized as eDNA from the environment accumulates on living organisms.

# INTRODUCTION

Every individual interacts with other individual organisms in some capacity, whether it be directly (*e.g.*, one species feeding on another) or indirectly (*e.g.*, nutrient transfer through secondary predation; *Fretwell, 1987*; *Boersma et al., 2008*). These interactions are essential to ecosystem functioning but can be challenging to identify. Yet, determining the nature of these interactions can provide insight into ecosystem processes, the structure of interaction

Corresponding author
Elizabeth L. Clare, eclare@yorku.ca

networks and, in the case of pests, may identify key agents of biocontrol. One possibility is to quantify these relationships by characterizing an organism's interactions using molecular tools through the amplification of multiple gene regions targeting different taxa. Interactions can be characterized in different ways. In ecological terms, an organism's "symbiome" refers to the collection of taxa (*i.e.*, bacteria, fungi, parasites, *etc.*) that interact with an individual of interest (*e.g.*, by residing in or on a host organism, or being eaten by it) and are both colocalized (co-occurring) and co-evolving (*Tripp et al., 2017*). The symbiome has been subtly differentiated from the holobiont which also includes physically associated taxa but not necessarily involving co-evolution. These concepts differ slightly from each other, and the symbiome may be a specific subset of the holobiont. The objective of these approaches is to characterize the interactions of a specific individual at a specific point in time, and differentiate this from more casual co-occurrences. This individual approach differs from concepts which consider repeated interactions (*e.g.*, dietary ecology; *Pompanon et al., 2012*) or pooling of data from individuals (*Drinkwater et al., 2019*), with species-level measurements as the unit of investigation. In practice, differentiating interactions which lead to co-evolution from those that are ecologically important, but do not lead to genomic changes, is difficult and it is likely that these interactions do not exist in discrete categories, but on a continuum from co-occurrence to co-evolution with gradually increasing interaction strength and selection profiles.

Many of these interactions result in the co-occurrence of DNA traces (*e.g.*, an herbivore would have the DNA of the plants it consumed in its stomach, but may also carry the DNA of species in the environment), which allows DNA barcoding and metabarcoding to be employed to identify these elusive relationships. The use of metabarcoding for the identification of ecological interactions relies on the amplification of multiple gene regions from the same DNA sample that vary significantly between species. For example, the mitochondrial cytochrome *c* oxidase subunit 1 (COI) can be used for animal identifications (*Hebert et al., 2003*), and the ITS region and rbcL gene can be used for genus level plant identifications (*CBOL Plant Working Group, 2009*). Sequencing generally employs high-throughput methods which allow for the identification of multiple taxa simultaneously from mixed samples (*Compson et al., 2020*). Metabarcoding can thus be used to characterize the biological interactions of ecologically or economically important species through the sequencing of multiple DNA target regions left behind by different interacting taxa. However, it is important to note that the interactions detected in this way should be considered "potential" interactions as DNA shed from organisms, which is ubiquitous in the environment (environmental DNA; eDNA), can make it difficult to differentiate a true interaction from the simple co-occurrence of DNA (see Discussion).

## Targeting insect ecological interactions

An ideal method for analysis would apply across a variety of animal targets of interest with minimal methodological variation. While a great many genes have been employed to diagnose species interactions (*Pornon et al., 2016*; *Rowe et al., 2021*; *Kim et al., 2022*), and some primers are designed specifically for this (*e.g.*, *Zeale et al., 2011*), fewer analyses have compared their use across different systems or hosts. Insects are an ideal model for

developing and testing target genes as their great diversity and ecological versatility offer a very wide spectrum of interactions (*Yang & Gratton, 2014*; *Crespo-Pérez et al., 2020*). In this work we developed a generalized method for the identification of potential ecological interactions by using multiple target genes for the recovery of interaction data from three insect groups of substantial economic importance in North America: the invasive spongy moth (*Lymantria dispar dispar*, Lepidoptera) and emerald ash borer (*Agrilus planipennis*, Coleoptera), as well as a variety of cavity nesting pollinators (Hymenoptera) in their larval form where taxonomic identity is challenging to establish.

## Spongy moths

Spongy moths were introduced to North America in 1869 and have caused substantial damage to more than 300 species of trees by defoliation during their caterpillar stage, reducing growth, root, and fruit production (*Elkinton & Liebhold, 1990*; *Liebhold et al., 1995*; *Davidson, Gottschalk & Johnson, 1999*; *Muzika & Liebhold, 1999*; *Kozlowski, Kramer & Pallardy, 1991*; *Nakajima, 2015*). Defoliated trees are often weaker and more susceptible to dying, with both economic and ecological impacts (*Davidson, Gottschalk & Johnson, 1999*). Defoliation reduces timber production (*Leuschner et al., 1996*), and in 2021 roughly 1.8 million hectares of forests in Ontario were defoliated by spongy moths (*Government of Ontario, 2014*). Between timber loss and biological controls, spongy moths cost North America nearly $3.2 billion annually (*Bradshaw et al., 2016*).

In addition to direct economic costs, spongy moth outbreaks cause disruption of ecological communities and food webs, reducing native moth populations and altering the structure and composition of forests (*Gurnell, 1983*; *Work & McCullough, 2000*; *Fajvan & Wood, 1996*; *Canham, 1988*). Efforts to control spongy moth populations and to prevent their spread include manually removing and killing the insects, the application of pesticides, and the introduction of biological controls (*i.e.*, natural enemies of the moths, such as the parasitic wasp *Ooencyrtus kuvanae* and the fungus *Entomophaga maimaiga*; *Mcmanus & Csóka, 2007*; *Elkinton & Liebhold, 1990*). Spongy moth populations fluctuate annually, with large-scale outbreaks occurring synchronously across large distances every 8 to 10 years, and smaller outbreaks every 4 to 5 years (*Berryman, 1996*; *Davidson, Gottschalk & Johnson, 1999*; *Peltonen et al., 2002*; *Haynes, Liebhold & Johnson, 2009*). The causes of these outbreaks have not yet been determined, but ecological interactions (*e.g.*, with predators and pathogens) have been hypothesized to play a role (*Allstadt et al., 2013*; *Elkinton & Liebhold, 1990*). As a result, determining large-scale species interactions may facilitate better predictions of outbreaks and present mechanisms for population control.

## Emerald ash borers

The emerald ash borer is a wood-boring beetle native to northeastern Asia. It was first detected in North America near Detroit, Michigan in 2002, although it may have been present undetected since the 1990s (*Cappaert et al., 2005*; *Siegert et al., 2014*). Emerald ash borer larvae consume the phloem and cambium of ash trees (*Fraxinus* spp.) creating S-shaped galleries that disrupt nutrient transport (*Cappaert et al., 2005*; *Wang et al., 2010*). Infestations are associated with extensive canopy dieback, especially in green ash (*Fraxinus*

*pennsylvanica*; *Anulewicz, McCullough & Cappaert, 2007*), causing tree mortality often exceeding 99% within 6–10 years of beetle arrival (*Natural Resources Canada, 2013*; *Knight, Brown & Long, 2013*; *Klooster et al., 2014*). As a direct result, six North American ash species have been added to the IUCN Red List of Threatened Species (*International Union for Conservation of Nature, 2022*). Since the beetle's introduction, tens of millions of trees have been lost (*Cappaert et al., 2005*) and they are estimated to be the costliest forest pest in North America, causing approximately $850 million in local government expenditures (*i.e.*, tree removal, replacement, and treatment) and $380 million in lost residential property values each year in the USA alone (*Aukema et al., 2011*).

Ecological impacts include reduced availability of food and shelter for organisms that rely on ash trees, *e.g.*, beavers, rabbits, deer, purple finches, and wood ducks (*Schlesinger, 1990*), and extinction risks in dependent species, *e.g.*, 43 monophagous native arthropod species like the eastern ash bark beetle (*Hylesinus aculeatus*) and black-headed ash sawfly (*Tethida barda*; *Gandhi & Herms, 2010a*). Like spongy moths, emerald ash borer infestations alter the structure and composition of forests: for example, canopy dieback allows increased light penetration to the subcanopy, encouraging the growth of understory species and altering their distribution and abundance (*Canham, 1988*; *Gandhi & Herms, 2010b*; *Flower, Knight & Gonzalez-Meler, 2013*). Identifying the broad-scale ecological interactions of emerald ash borers may allow for the monitoring of known biological controls or for the identification of novel ones.

## Pollinators

Bees are arguably the best-known pollinators, and eusocial bees are heavily managed for efficient, mass agricultural pollination (*Kleijn et al., 2015*; *Klein et al., 2006*), due to their domesticity, ease of transport, and large numbers (*Iwasaki & Hogendoorn, 2021*). As they offer key ecosystem services such as agricultural pollination and, in the case of the honeybee (*Apis mellifera*), production of honey, they have been a major focus for conservation (*Iwasaki & Hogendoorn, 2021*). Despite the popularity of these commercial pollinators, most pollinators (bees, wasps, other insects) are not colonial and cannot be managed commercially yet provide crucial pollination services in both natural and agricultural systems (*Garibaldi et al., 2011*, *2013*; *Park et al., 2016*; *Russo et al., 2015*). Because they are not colonial, these taxa are more challenging to study. They can be collected in reasonably large numbers as larva but then there is an increased challenge related to their taxonomic identification. Increasing evidence suggests that non-colonial pollinator contributions to agricultural pollination are underestimated and their ecological interactions differ from colonial honeybees requiring alternative management strategies. For example, *Gresty et al. (2018)* demonstrated that floral provision thought to bolster pollinator populations and mitigate the effects of large-scale agricultural land conversion were not well used by solitary bees, suggesting different ecological interactions. Significant behavioral differences (*e.g.*, shorter flight seasons in native bees; *Bosch & Kemp, 2002*) and regional differences in community composition (*Danforth, Minckley & Neff, 2019*) make it difficult to extrapolate ecological data across taxa and region. Analysis of symbiont and holobiont data could provide a rapid mechanism to evaluate ecological interactions where

visual observations are not practical, particularly in low density populations with short duration seasonal activity.

In this study we examine potential target regions and protocols to develop a general metabarcoding approach which can be applied to a variety of insects to detect their potential interactions with other taxa. Our objectives are to: (1) evaluate different target regions for the identification of potential ecological interactions in these taxonomic groups, (2) assess whether these analyses should be targeted at different life stages (*e.g.*, eggs, larvae, pupae, and adults), and (3) compare the results of "pooling" polymerase chain reaction (PCR) products from different gene regions into a single sequencing library *vs.* individual analysis for each marker for analytical efficiency. Direct comparisons of the taxonomic coverage of target gene regions for metabarcoding of potential interactions across a range of insect models will guide methodological choices in ecological analyses.

## MATERIALS AND METHODS

### Sample collection and study design

All samples used in this analysis were acquired from the collections of the Centre for Biodiversity Genomics, University of Guelph, Canada. They had been collected and sorted using clean protocols (gloves, cleaned tools, *etc.*) to minimize any cross contamination and stored in individual sealed vials. To simplify methodological testing, we developed our protocol using a hierarchical testing design. We started with spongy moths as pooled samples following the approach for *Drinkwater et al. (2019)*. While this does not assess individual interactions, we used this approach to increase DNA yield for initial primer testing. We tested a subset of primers on these samples. We then used an individual approach for a large set of emerald ash borer individuals (where a sample consisted of a whole or partial individual), and larvae from homes deployed to attract native and non-colonial bees, while also reducing reaction volumes and increasing the number of target regions.

#### Spongy moths

We acquired 18 samples of spongy moth eggs and larvae. A sample consisted of a pool of individuals from the same tree to increase DNA yield (*Drinkwater et al., 2019*; see below and Table 1). These samples had been hand-collected from the University of Guelph Arboretum (Guelph, Ontario, Canada) in May 2021 from a variety of host trees, including maple ($n = 1$), larch ($n = 1$), poplar ($n = 1$), birch ($n = 2$), pine ($n = 2$), beech ($n = 1$), and cherry ($n = 1$). The specimens were stored in 95% ethanol and kept at $-20\ °C$.

#### Emerald ash borers

We acquired 277 samples of emerald ash borer larvae (three sizes hereafter larva size 1, larva size 2, larva size 3), pupae, and imagoes. These samples had been hand-collected from ash trees found in various locations across Brockville ($n = 185$), Ennotville ($n = 41$), Guelph ($n = 41$), and Puslinch ($n = 10$) in Ontario, Canada. Of the 277 specimens, there were 60 imagoes (six were known to be parasitized/fungal infested), 53 pupae, 54 size 1 larvae (<1.3 cm; 11 parasitized/fungal infested), 68 size 2 larvae (1.3–2.5 cm; 21 parasitized/fungal infested), 41 size 3 larvae (>2.5 cm; one parasitized/fungal infested), and one specimen of

**Table 1 Target regions for analysis of insect samples including *Lymantria dispar dispar* (spongy moth), *Agrilus planipennis* (emerald ash borer), and larva from native pollinators.** All primers were subsequently modified with CS1/CS2 tails for sequencing using the Illumina MiSeq platform with the exception of SSU_F_07/SSU_R_26 which were modified following *Hebert et al. (2025)* for nanopore sequencing.

| Gene | Primer | Reference | Direction | Primer sequence (5′ to 3′) | Amplicon length (bp) | Insect test samples |
|---|---|---|---|---|---|---|
| COI | ZBJ-ArtF1c | *Zeale et al. (2011)* | Forward | AGATATTGGAACWTTATATTTTATTTTTGG | 157 | LDD: nine egg masses, nine larvae EAB: 57 adults, 53 pupae, 54 size one larvae, 68 size two larvae, 41 size three larvae, one unknown |
| | ZBJ-ArtR2c | *Zeale et al. (2011)* | Reverse | WACTAATCAATTWCCAAATCCTCC | | |
| COI | BF3 | *Elbrecht et al. (2019)* | Forward | CCHGAYATRGCHTTYCCHCG | 418 | Pollinators: 71 larvae |
| | BR2 | *Elbrecht & Leese (2017)* | Reverse | TCDGGRTGNCCRAARAAYCA | | |
| rbcL | rbcl_1–8 | *Palmieri, Bozza & Giongo (2009)* | Forward | TTGGCAGCATTYCGAGTAACTCC | 226 | LDD: nine egg masses, nine larvae Pollinators: 71 larvae |
| | rbcLB | *Palmieri, Bozza & Giongo (2009)* | Reverse | AACCYTCTTCAAAAAGGTC | | |
| ITS | ITS-S2F | *Chen et al. (2010)* | Forward | ATGCGATACTTGGTGTGAAT | ~300–400 | LDD: nine egg masses, nine larvae Pollinators: 71 larvae |
| | ITS4 | *White et al. (1990)* | Reverse | TCCTCCGCTTATTGATATGC | | |
| ITS | ITS3 | *White et al. (1990)* | Forward | GCATCGATGAAGAACGCAGC | ~300–400 | EAB: 60 adults, 53 pupae, 54 size one larvae, 68 size two larvae, 41 size three larvae, one unknown specimen |
| | ITS4 | *White et al. (1990)* | Reverse | TCCTCCGCTTATTGATATGC | | |
| 16S | 799F-mod3 | *Hanshew et al. (2013)* | Forward | CMGGATTAGATACCCKGG | ~300–400 | LDD: nine egg masses, nine larvae EAB: 57 adults, 53 pupae, 54 size one larvae, 68 size two larvae, 41 size three larvae, one unknown Pollinators: 71 larvae |
| | 1115R | *Reysenbach & Pace (1995)* | Reverse | AGGGTTGCGCTCGTTG | | |
| 18S | SSU_F_07 | *Floyd et al. (2002), Carta & Li (2018)* | Forward | AAAGATTAAGCCATGCATG | ~1,000 | EAB: 57 adults, 53 pupae, 54 size one larvae, 68 size two larvae, 41 size three larvae, one unknown specimen |
| | SSU_R_26 | *Floyd et al. (2002), Carta & Li (2018)* | Reverse | CATTCTTGGCAAATGCTTTCG | | |

an unknown life stage with an observed symbiont worm. We specifically included individuals with observed fungal or parasite interactions to increase the complexity for amplicon targeting, and as a positive control for taxa detection. The specimens were stored in 95% ethanol and kept at −20 °C.

### Pollinators

We acquired 71 genomic extracts from larvae collected across Canada from bee homes set up at schools in the summer of 2019 and 2020. Each extract consisted of multiple larvae pooled with DNA extracted using a NucleoMag Plant Kit (Macherey NagelDüren, Düren, Germany). They had been previously screened for ID and the samples used here contained only larvae confirmed to be pooled from the same species (number of pooled larva varied) and available as an archived DNA sample for analysis. Full protocols for these larva are described in *Handler et al. (2024)*.

## DNA extraction

For both spongy moths and emerald ash borers we extracted DNA using the Qiagen blood and tissue kit following the manufacturer's guidelines, with two modifications. First, specimens were ground using a pestle during initial lysis, and second, we decreased the final DNA elution to 100 μL of Buffer AE for spongy moths and 150 μL for emerald ash borers, to increase DNA yields. For spongy moth extractions we used 4–5 eggs or larvae from the same host tree in order to get sufficient biomass to extract DNA, thus each "sample" is a pool of 4–5 individuals (*e.g.*, see the method used by *Drinkwater et al., 2019*). For emerald ash borers we extracted DNA from the posterior half of the abdomen for imagoes, the posterior third for pupa specimens, and an approximately 6 mm fragment of the posterior end of larva specimens (or the whole specimen if classified as larva size 1). Extraction blanks were included using sterile water.

## Amplification of target gene regions

### Spongy moths

A total of 80 PCRs were generated using multiple primers to amplify different taxonomic targets (animals, plants, microbes, and fungi) within each sample. Positive controls (cockroach and its microbiome for COI and 16S, and cannabis for ITS and rbcL) and negative controls (water) were included with each round of amplification. All primers had been modified with CS1/CS2 tails (see modifications described by *Ison et al., 2016*) for sequencing using the Illumina MiSeq platform and diluted to 10 μM. One PCR was generated per sample and per target region (18 samples and two controls × four regions).

**rbcL plant amplification (primers rbcl_1–8/rbcLB; *Palmieri, Bozza & Giongo, 2009*) and COI insect amplification (primers ZBJ-ArtF1c/ZBJ-ArtR2c; *Zeale et al., 2011*):** Each 20 μL reaction contained 12.5 μL Qiagen multiplex PCR master mix, 4.5 μL water, 1.25 μL of each primer, and 0.5 μL DNA template (or water for the negative control). The PCR thermocycling conditions were as follows: 5 min at 95 °C, followed by 35 cycles of 30 s at 95 °C, 45 s at 50 °C, and 50 s at 72 °C, followed by 5 min at 72 °C (adapted from

*Elbrecht et al., 2019*). The same protocol was used for the COI primers, except only 25 cycles were run.

**ITS plant and fungal amplification (primers ITS-S2F/ITS4;** *Chen et al., 2010*; *White et al., 1990*): Each 25 μL reaction contained 12.5 μL Qiagen multiplex PCR master mix, 9 μL water, 1.25 μL of each primer, and 1 μL DNA template (or water for the negative control). The PCR thermocycling conditions were as follows: 4 min at 94 °C, followed by 34 cycles of 30 s at 94 °C, 40 s at 55 °C, and 1 min at 72 °C, followed by 10 min at 72 °C (*Cheng et al., 2016*).

**16S microbial amplification (primers 799F-mod3/1115R;** *Hanshew et al., 2013*; *Reysenbach & Pace, 1995*): Each 20 μL reaction contained 12.5 μL Qiagen multiplex PCR master mix, 5 μL water, 0.75 μL of each primer, and 1 μL DNA template (or water for the negative control). The PCR thermocycling conditions were as follows: 2 min at 95 °C, followed by 35 cycles of 20 s at 95 °C, 30 s at 48 °C, and 30 s at 72 °C, followed by 3 min at 72 °C (adapted from *Hanshew et al., 2013*).

### Emerald ash borers

A total of 1,129 PCRs were generated using multiple primers to amplify the same target lineages (animals, plants, bacteria, and fungi) within the emerald ash borer samples. Extraction controls and positive (cricket for COI, pepper for ITS and 16S) and negative (water) PCR controls were included with each amplification. Based on amplification success rates in spongy moths we modified our choice of primers (see Table 1), reaction volumes, and amplification protocols to reduce costs and simplify the analysis for broader application in ecological analysis. In short, we excluded plant identifying primers (rbcl_1–8/rbcLB and ITS-S2F/ITS4) since emerald ash borers spend much of their life inside of ash trees and are not expected to interact with any other plants. To ensure fungi were still targeted, we included the ITS3/ITS4 primer pair. We also evaluated a variety of 18S regions to include parasite interactions. One PCR was generated per sample and per target region, for a total of 283 PCRs generated using the COI region (including six controls and three extraction blanks), 285 with the ITS region (including five controls and three extraction blanks), 283 with the 16S region (including six controls and three extraction blanks), and 278 with the 18S region (including three controls and three extraction blanks). There were three sample DNA extracts that were accidentally destroyed before we could generate PCRs for the COI, 16S, and 18S regions.

**COI insect (primers ZBJ-ArtF1c/ZBJ-ArtR2c;** *Zeale et al., 2011*), **ITS plant and fungal (primers ITS3/ITS4;** *White et al., 1990*), **and 16S microbial (primers 799F-mod3/1115R;** *Hanshew et al., 2013*; *Reysenbach & Pace, 1995*) **amplification:** Each 15 μL reaction contained 7.5 μl Qiagen multiplex PCR master mix, 4.5 μL water, 1 μL of each primer, and 1 μL DNA template (or water for the negative control). The PCR thermocycling conditions were as follows: 5 min at 95 °C, followed by 35 cycles of 30 s at 94 °C, 45 s at 52 °C, and 1 min at 72 °C, followed by 10 min at 72 °C. All primers had been modified with CS1/CS2 tails for sequencing using the Illumina MiSeq platform.

**18S parasite amplification (primers SSU_F_07/SSU_R_26, formerly SSU18A and SSU26R;** *Floyd et al., 2002*; *Carta & Li, 2018*): 18S targets present a tradeoff between

shorter reads with only higher-level taxonomic assignments, or longer reads with better taxonomic resolution but incompatible with most high throughput sequencing platforms. We evaluated several 18S primers (563F/1132R, 1391F/EukBr, and some in house designs) but amplification was most successful for long read amplicons which cannot easily be used for the most common high-throughput sequencing (HTS) platforms. For emerald ash borers we used the protocols for high throughput barcoding on the MinION sequencing platform (Oxford Nanopore Technologies, Oxford, UK) with primer modifications as described in *Hebert et al. (2025)*. Each 12.5 μL reaction contained 6.25 μL 10% trehalose (Fluka Analytical), 1.25 μL 10× PlatinumTaq buffer (Thermo Scientific, Waltham, MA, USA), 1.62 μL water, 0.625 μL of 2 μM forward primer, 0.625 μL of 100 μM reverse primer, 0.625 μL 50 nM $MgCL_2$, 0.0625 μL of 10 nM dNTPs (Kapa Biosystems, Wilmington, MA, USA), 0.06 μl of 5 U/μL PlatinumTaq (Thermo Scientific, Waltham, MA, USA), and 2 μL DNA template. The PCR thermocycling conditions were as follows: 2 min at 94 °C, followed by 35 cycles of 40 s at 94 °C, 40 s at 56 °C, and 1 min at 72 °C, followed by 5 min at 72 °C.

### Pollinators

A total of 290 PCRs were generated using multiple primers to amplify target lineages (insects, plants, bacteria, fungi) within larvae. Extraction controls and positive (cricket for COI, pepper for ITS and 16S) and negative (water) PCR controls were included with each amplification. Because we expect a larger range of plant interactions (*e.g.*, not confined to one host plant) we included rbcL as a marker to increase taxonomic coverage. As bee homes are frequently used by multiple species and subject to nest parasitism by other insects, and larvae are extremely challenging to identify morphologically, we included a COI insect target to provide both larval host and parasitoid identification. Each larval sample was used to generate one PCR per target region, with 72 PCRs generated in the COI region (one control), 73 with the rbcL region (two controls), 73 with the ITS region (two controls), and 72 with the 16S region (one control).

**COI Hymenoptera (primers BF3/BR2; *Elbrecht & Leese, 2017*; *Elbrecht et al., 2019*) and rbcL plant amplification (primers rbcL1/rbcLB; *Palmieri, Bozza & Giongo, 2009*):** Each 20 μL PCR reaction contained 12.5 μL Qiagen multiplex PCR master mix, 4.5 μL water, 1.25 μl of each primer, and 0.5 μL template DNA. The PCR thermocycling conditions were as follows: 5 min at 95 °C, followed by 25 cycles of 30 s at 95 °C, 45 s at 50 °C and 50 s at 72 °C, followed by 5 min at 72 °C (*Elbrecht et al., 2019*; *Little, 2014*; *Roger et al., 2022*). Cycles were increased to 30 for rbcL.

**ITS plant and fungal amplification (primers ITS-S2F/ITS4; *Chen et al., 2010*; *White et al., 1990*):** Each 25 μL PCR reaction contained 12.5 μL Qiagen multiplex PCR master mix, 9 μL water, 1.25 μL of each primer, and 1 μL template DNA. The PCR thermocycling conditions were as follows: 4 min at 94 °C, followed by 34 cycles of 30 s at 94 °C, 40 s at 55 °C and 1 min at 72 °C, followed by 10 min at 72 °C (*Cheng et al., 2016*).

**16S microbial amplification (primers 799F-mod3/1115R; *Hanshew et al., 2013*; *Reysenbach & Pace, 1995*):** Each 20 μL PCR reaction contained 12.5 μL Qiagen multiplex PCR master mix, 5 μL water, 0.75 μL of each primer, and 1 μL template DNA. The PCR

thermocycling conditions were as follows: 2 min at 95 °C, followed by 35 cycles of 20 s at 95 °C, 30 s at 48 °C and 30 s at 72 °C, followed by 3 min at 72 °C (*Rothman et al., 2019*).

### Pooled vs. individual PCRs from pollinators

There is a cost in both time and money to independently barcode and process PCR templates for sequencing. Pooling PCRs from different markers could reduce costs. To test the impact of independent processing *vs.* pooled processing of PCR products from different markers we prepared pollinator PCR reactions in two ways. First, each PCR from each sample was independently processed (see above). Second, a 2 μL aliquot of each PCR product for each sample was pooled and mixed (*i.e.*, 2 μL each of COI, rbcL, ITS, and 16S PCR product generated using the same template DNA was mixed to make a new "Pooled PCR" for this template). This resulted in 72 pooled samples in total (one per pollinator sample and one negative control). Independent and pooled templates were processed in the same sequencing batch so that the only difference in protocol was the pooling (same PCR templates, same barcoding and sequencing protocol).

## Sequencing and data analysis

Amplified DNA was sent to the Barts and the London Genome Center at Queen Mary University of London's Blizzard Institute, with the exception of 18S fragments (see below). All samples sent were processed identically including bead clean up and size selection by 0.9x Ampure Beads (Beckman Coulter, Brea, CA, USA), quantification, quality control (QC) and normalization using Qubit (Invitrogen, Carlsbad, CA, USA) nucleotide quantification, and DNA D100 Tape station (Agilent). All PCRs were independently barcoded using single indexes and sequenced using a MiSeq v3 2 × 300 cycle run (Illumina, San Diego, CA, USA). Raw sequences were demultiplexed on site.

Cutadapt v3.7 (*Martin, 2011*) was used in paired end mode to separate markers in pooled libraries based on primer pair. Then, these were processed the same way as their individual PCRs correspondents. Demultiplexed ITS and rbcL files were processed using the DADA2 pipeline (*Callahan et al., 2016*) in R Studio (*R Core Team, 2023*) following *Garrett et al. (2023)*. We first removed amplicon primers using Cutadapt. In brief, DADA2 was used to merge paired reads, trim sequence length, and remove errors in sequence profiles using the *learnErrors* function. Chimeras were removed and amplicon sequence variant (ASV) tables were generated for each primer pair and each taxonomic group individually. Resulting ASVs for ITS and rbcL sequences were compared to the nucleotide collection in GenBank using the BLAST algorithm. An identification was retained for further consideration if the identity match was ≥97% (with 100% overlap), and if the detection count was greater than the highest read count in the negative control (negative filtering). We then screened taxonomic identity against available ecological data on distributions (*i.e.*, the organism can be found in the area the samples were collected in according to the Global Biodiversity Information Facility (*Global Biodiversity Information Facility (GBIF), 2023*). Several potential interactions of specific interests which met some but not all of these conservative criteria are also reported but with cautionary notes (see Results).

16S reads were processed using QIIME2 (*Bolyen et al., 2019*). Sequences were trimmed using Cutadapt v3.7 (*Martin, 2011*) and denoised using DADA2 (*Callahan et al., 2016*). The denoising parameters (trim-left-f, trim-left-r, trunc-len-f, and trunc-len-r) were as follows: 0, 0, 268, and 218 for emerald ash borers; 0, 0, 271, and 202 for spongy moths; 0, 0, 272, and 203 for pollinators. An identification was retained if the detection count was greater than the highest read count in the negative control, if it was not associated with the positive control, and if the taxonomic identity matched known ecological data (*i.e.*, the organism is commonly associated with soil/water/plants or is an insect pathogen according to MicrobeAtlas v1.0 (*Matias Rodrigues et al., 2017*) and is thus unlikely to be a human/lab contaminant).

Spongy moths and emerald ash borer COI sequences were processed in mBRAVE (*Ratnasingham, 2019*) using the default parameters except for the following: trim front = 30 bp, trim end = 24 bp, min Quality Value (QV) = 0 qv, max length (pre-trim) = 600 bp, max bases with low QV (<20) = 75%, max bases with ultra-low QV (<10) = 75%, ID distance threshold = 5%, exclude from operational taxonomic unit (OTU) threshold = 5%, read sub-sampling-max reads per sample = 2,500, read sub-sampling-max reads per contig = 200, paired end merging = merge, assembler min overlap = 10 bp, and assembler max substitution = 20 bp. The following system reference libraries were used: Insecta (SYS-CRLINSECTA), Non-Insect Arthropoda (SYS-CRLNONINSECTARTH), Non-Arthropoda Invertebrates (SYS-CRLNONARTHINVERT), and Chordata (SYS-CRLCHORDATA). Pollinator COI was similarly processed but used the CBG Authoritative Canadian Reference Library 2022 (DS-CANREF22) with trim front and end set to 20, ID distance threshold = 1.5%, exclude from OTU threshold = 3%, read sub-sampling – max reads per sample = 2,000, min QV = 10 qv, max length = 1,000 bp, and assembler max substitution = 5 bp. An identification was retained for further evaluation if the mean length fell within the expected range (*i.e.*, expected amplicon length ± 1 bp), if "pc_sim_mean" was ≥97%, and if "id_gaps_mean" equaled zero. We used full negative filtering using the negative controls. All putative taxonomic identifications were compared against known distribution records (*e.g.*, in *Global Biodiversity Information Facility (GBIF), 2023*). The coleopterans *Tylonotus bimaculatus* (Cerambicidae) and *Hylesinus aculeatus* (Curculionidae) were detected in some presumed emerald ash borer samples, and these were removed from further analysis as larvae were likely misidentified during the collection stage. For each larval sample, the ASV with the highest read count above 100 reads was considered the "occupant" of the sample (*Handler et al., 2024*). If all read counts were lower than 100, or there were two very similar IDs it was labelled as an "unclear ID" sample reflecting poor amplification or ID in general. All other associated ASVs were considered potential interactions. The most common occupants (found in a minimum of three samples) were chosen as a focus for analysis.

Because the 18S amplicons exceed the length which can be processed on Illumina platforms (>900 bp for the target nematodes), 18S emerald ash borer amplicons were sequenced using the MinION platform (Oxford Nanopore, Oxford, UK) with library prep carried out as described in *Floyd, Prosser & Jafarpour (2023)* and run on a Flongle flow cell. Data was analyzed using the methods detailed in *Hebert et al. (2025)*. To assign putative
taxonomic identification to sequences, the SINTAX tool (*Edgar, 2016*) was employed, with a custom reference library containing the nematode sequences from 18S NemaBase (*Gattoni et al., 2023*) and sequences of all other taxa from the SILVA public database.

All figures were generated in R (*R Core Team, 2023*) using the ggplot2 package (*Wickham, 2016*) and arranged into multi-panel figures in power point.

## RESULTS

### Spongy moths

We processed 1,460,234 COI, 199,979 ITS, 539,627 rbcL and 292,782 16S raw reads. These were reduced to 281 non-host reads (spongy moth data was excluded), 92,616, 377,885 and 18,091 filtered reads which were converted to 15, 167, 42 and 97 BINS (COI) or ASVs respectively. We identified 126 taxa across the four markers (ITS = 49; 16S = 41; COI = 8; rbcL = 28). Of these, 119 were unique taxa (plants identified by both rbcL and ITS were not counted twice). Most identifications were made to level of genus, and these represented 60 different orders, with two unclassified algae and an uncultured microbe included in this count. We identified 19 orders (six fungi and 13 plants) using the ITS marker, detecting 17 in the eggs and 14 in the larvae (Table S1). Using the 16S marker we identified 28 bacteria orders, with 22 and 20 orders detected in the egg and larvae life stages, respectively (Table S2). We detected *Wolbachia* sp. (order Rickettsiales), an insect symbiont, in five samples. Using the COI marker, we detected six orders of insects, other than the host itself, with five orders in the egg stage and four in the larval stage (Table S3), which we suspect represent potential parasites. We detected 15 plant orders with rbcL (eight of which were also detected using ITS), with eggs containing plant DNA from 13 orders, and larval samples containing plant DNA from 11 (Table S4).

These detections from 60 orders accounted for 317 actual occurrences across the two life stages (Fig. 1). There were 167 order-level occurrences in the egg life stage (nine animal, 60 bacteria, 21 fungi, and 77 plant), while we recorded 150 occurrences in the egg life stage (six animal, 64 bacteria, 14 fungi, and 66 plant).

### Emerald ash borers

We processed 11,361,252 COI, 2,443,232 ITS, 6,173,341 16S, and 287,180 18S raw reads. These were reduced to 114,088 non-host reads (emerald ash borer data was excluded), 942,740, 8,154 and 8,126 filtered reads which were converted to 6, 36, 38, and 31 BINS (COI, 18S) or ASVs respectively. We identified 65 taxa across the four markers (ITS = 23; 16S = 21; COI = 4; 18S = 17). These represented 61 unique identifications from 30 orders. We detected 11 orders of fungi using the ITS marker, with larva size 1 showing a richness of five orders, larva size 2 seven orders, larva size 3 three orders, pupae six orders, and imagoes eight orders (Table S5). Notably, we detected the fungal biocontrol agent *Beauveria* sp. (order Hypocreales) in two samples. However, it should be noted that this was also detected in one of the negative controls. We retained the occurrence here because it was present in very small amounts in this control compared those detected in the samples and there is no other known source of the fungus in our processing. It was also independently verified by 18S data (see below) supporting it as a true positive. Using the

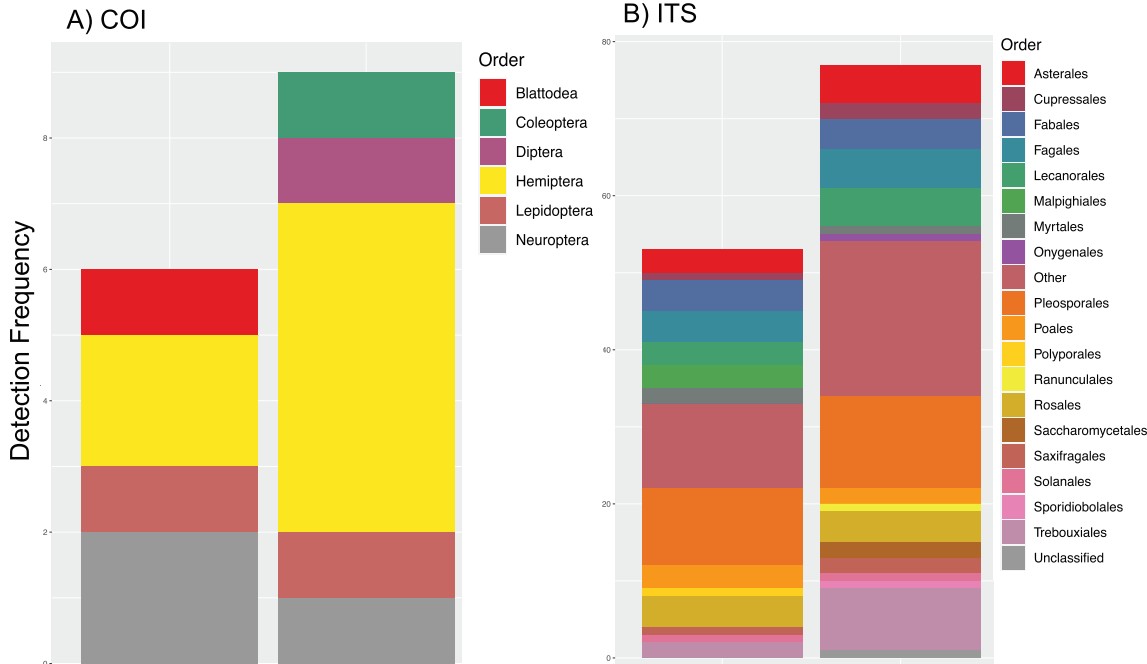

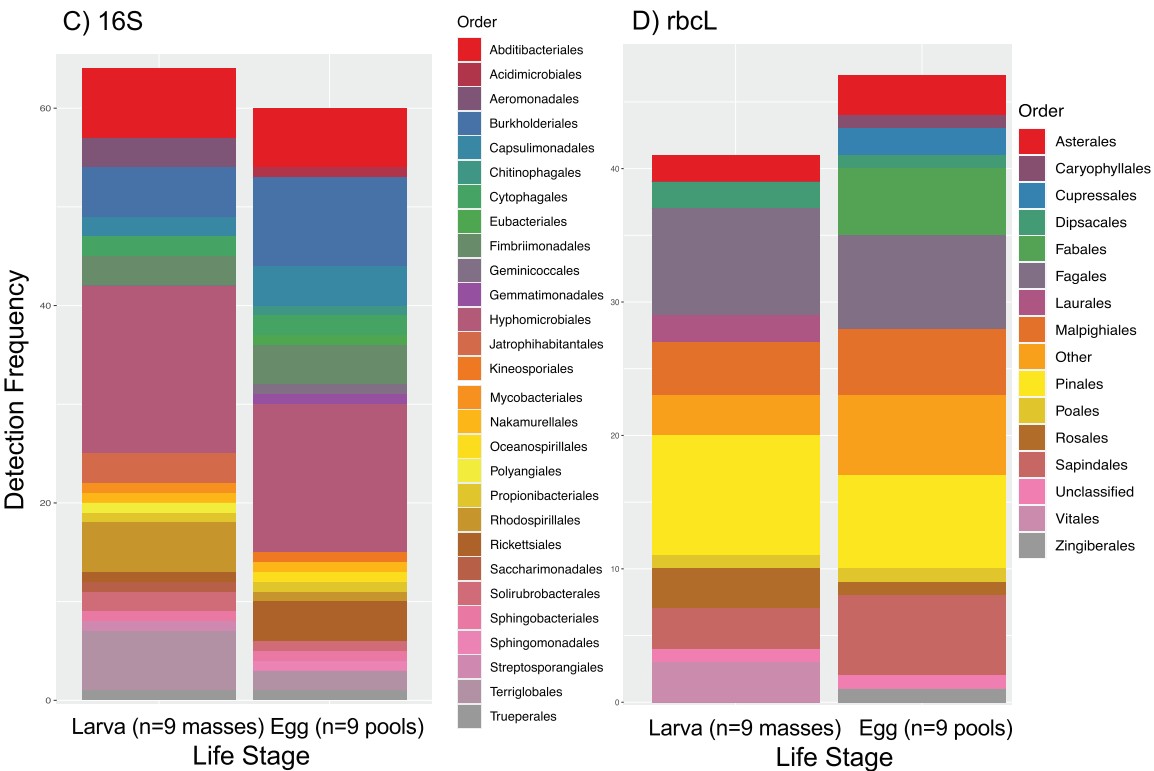

**Figure 1 Co-amplified taxa from host spongy moths.** (A) In addition to the amplification of host DNA, COI primers co-amplified DNA from six orders of insects as minor background signals. (B) ITS primers recovered 19 orders of plant and fungi in the DNA samples of moths with more taxa detected in egg masses than larval samples. (C) 16S primers targeting the microbiome detected a remarkably similar microbial community in egg *vs.* larval stages. (D) rbcL primers targeting plant DNA detected a similar, but not completely overlapping plant community to ITS primers (B) with the same increased taxonomic richness in egg masses compared to larval samples.

16S marker, we detected 13 orders of bacteria. We identified five different orders in the larva size 1 stage, three in the larva size 2 stage, two in the larva size 3 stage, eight in the pupal stage, and nine in the adult stage (Table S6). We detected the insect symbiont *Wolbachia* sp. (order Rickettsiales) in one sample. We also detected four arthropod orders using the COI marker, including potential parasites. Two orders were detected in the larva size 2, one in the larva size 3 and pupal stage, and two in the adult stage. Notable identifications included Ichneumonidae parasitoid wasps (Table S7). With the 18S marker, nine orders were detected, with six of the nine representing fungi (all overlapping with those identified using ITS), and the remaining three representing insects, nematodes, and a parasitic alveolate protist (*Cryptosporidium* sp., order Cryptosporida): three in the larva size 1 stage, five in the larva size 2, one in the larva size 3, four in the pupae, and two in the imagoes (Table S8).

These detections from 30 orders accounted for 259 actual detections across life stages (Fig. 2). There were 29 potential order level interactions detected in larva size 1 (three animal, 17 bacteria, and nine fungi), 64 in larva size 2 (eight animal, 24 bacteria, and 32 fungi), 10 in larva size 3 (two animal, three bacteria, and five fungi), 74 in the pupae (five animal, 31 bacteria, and 38 fungi), and 82 in the adult beetles (four animal, 48 bacteria, 29 fungi, and one parasitic alveolate).

## Pollinators

We processed 2,054,675 COI, 1,395,445 ITS, 1,649,988 rbcL, and 2,731,151 16S raw reads. These were reduced to 221,846, 1,088,989, 1,564,079 and 2,220,678 filtered reads which were converted to 188, 1,538, 727 and 3,169 BINS (COI) or ASVs respectively. We identified 330 taxa from 89 orders across the four markers (rbcL = 22; ITS = 24 plants and 15 fungi; 16S = 20; COI = 8 Tables S9–S12). These represented 73 unique order-level identifications. A total of 16 orders of plants were identified by both rbcL and ITS and were not counted twice. We identified 24 species of Hymenoptera among the larvae, the most common of which were the bees *Heriades carinatus*, *Megachile campanulae*, *M. centuncularis*, *M. relativa*, *Osmia tersula*, and the wasp *Symmorphus bifasciatus*. Other COI identifications included known parasites and prey items (*e.g.*, of the wasp *Apocrita* sp. which also uses solitary bee homes). We identified 17 fungal genera of which the most common was *Alternaria*, followed by *Ascosphaera*, *Aspergillus*, and *Penicillium*. There was overlap between plants identified using ITS and rbcL but much higher richness estimates from ITS (*e.g.*, all individuals had more than 20 genera of plant identified with ITS while this was a maximum identified with rbcL). The most common plant genera identified were associated with the Asteraceae, Brassicaceae, Fabaceae, and Rosaceae. Using the 16S region we identified 39 bacterial genera, the most common being *Wolbachia*, followed by *Apilactobacillus* and *Mellisococcus*.

## Individual vs. pooled samples from pollinators

Given that the same PCRs were used for pooled and individual libraries, the similarity was surprisingly small between them. Taxonomic coverage for pooled PCRs was much lower than individual PCRs in all cases (Fig. 3). The best coverage for pooled PCRs was offered by

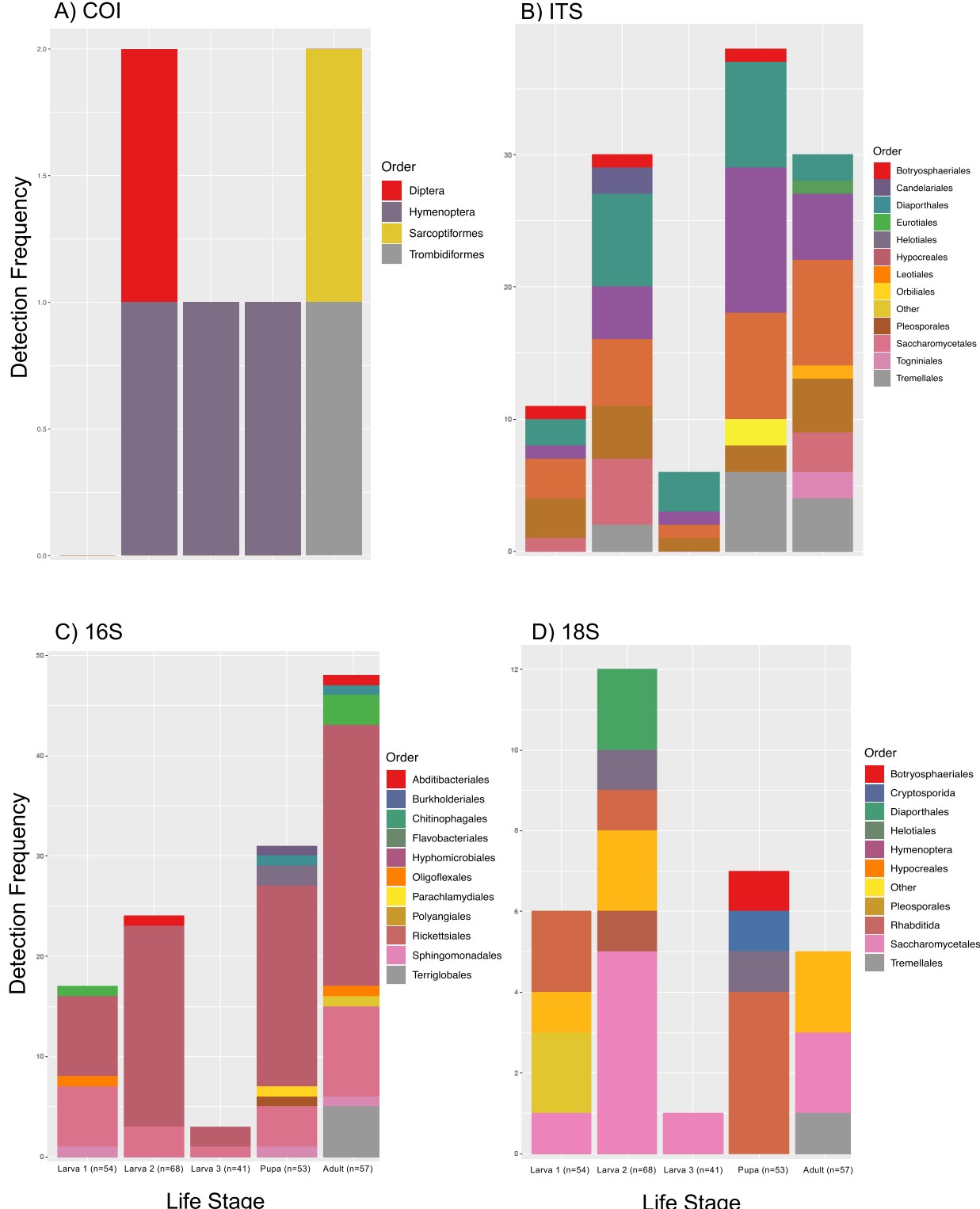

**Figure 2 Co-amplified taxa from host emerald ash borer across life stages.** (A) In addition to the amplification of host DNA, COI primers co-amplified four orders of insects as minor background signals. (B) ITS primers targeting primarily fungal DNA recovered 11 orders in the DNA samples of larvae, pupae, and adults. (C) 16S primers targeting the microbiome found an increasingly complex microbial community with life stage. (D) 18S primers amplified the widest range of taxa including a series of nematodes which are likely parasites. Low taxonomic recovery from larva size 3 specimens likely reflects the small sample size.

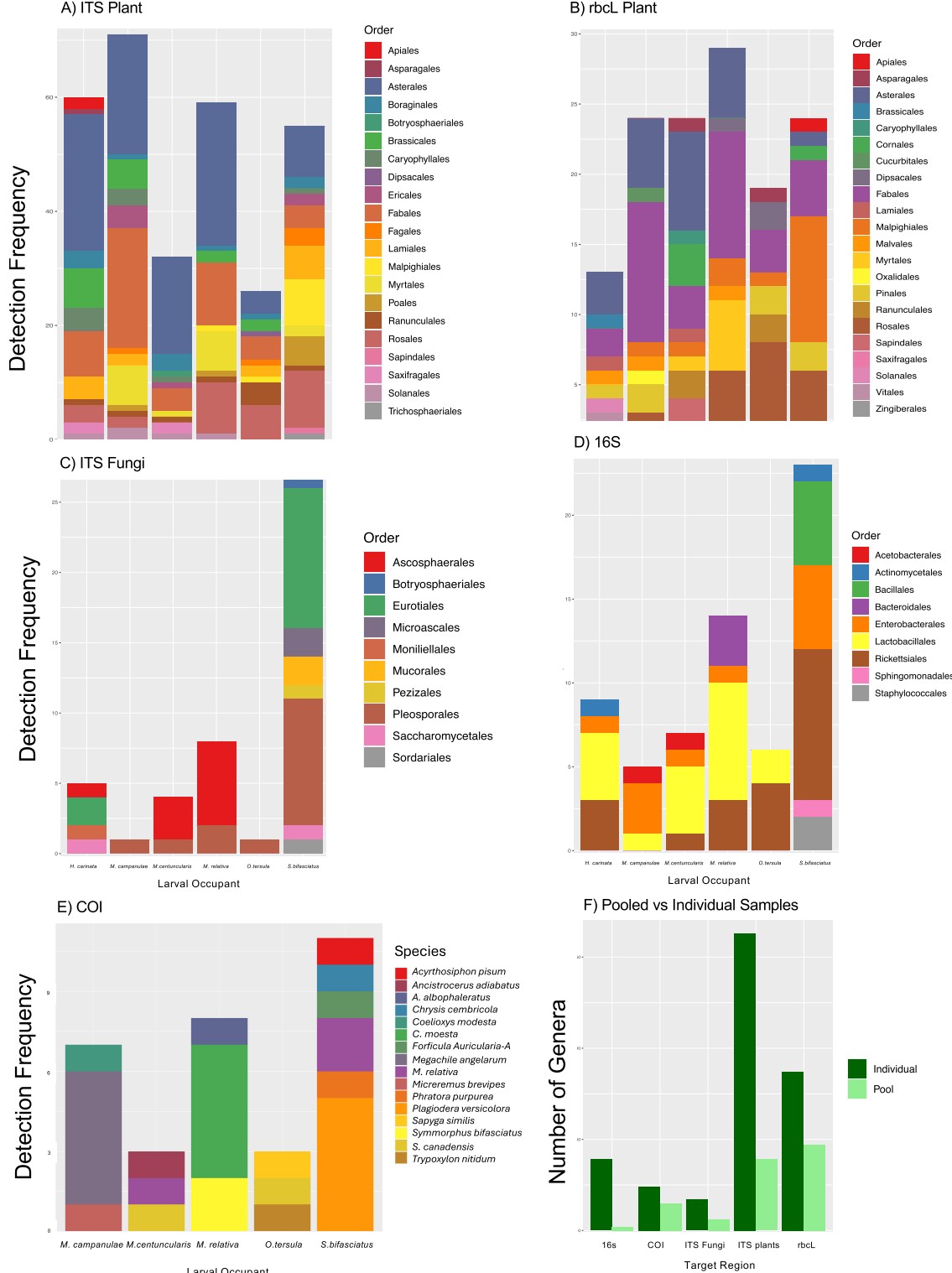

**Figure 3  Co-amplified taxa from pollinator larva in artificial bee homes.** (A) ITS primers targeting primarily plant DNA recovered 21 orders in the DNA samples of larva collected from tubes in bee homes; (B) rcbL primers targeting plant DNA detected 22 orders of plants of which 14 were also detected by ITS; (C) ITS primers targeting primarily fungal DNA detected 10 orders; (D) 16S primers targeting the microbiome recovered nine orders of bacteria; (E) in addition to the amplification of host DNA COI primers co-amplified 16 additional arthropods including known

**Figure 3 (continued)**
kleptoparasites, prey, and some evidence of cross amplification from other neighbouring larva (*e.g.*, *S. bifasciatus* is a host but also detected in *M. relativa* larva); (F) a greater number of genera were recovered from individual samples than pooled samples across all primers tested.

the COI marker, where 15 genera were detected in pooled PCRs *vs.* 24 in the same samples when not pooled. Comparatively, the other markers had more notable differences in coverage. Individual ITS plant and fungal amplifications identified 180 genera but only 45 were detected in the pooled samples with one taxa, *Parthenocissus* unique to pooled data. Within the 16S data, two genera were identified in pooled samples compared to the 39 identified in individual samples. Pooled COI data recovered far fewer host larval IDs, with most not meeting the criteria for identification of a host. The identified genera for the pooled samples frequently did not include most common genera in the individual samples.

## DISCUSSION

In this work we compare the performance of several molecular markers for the identification of potential cross-species interactions. We used seven primer pairs from five gene regions targeting insects, fungi, plants, bacteria, and parasites to detect potential ecological interactions of commercially destructive spongy moths and emerald ash borers and a variety of non-colonial pollinators. Our objective was to develop a generalizable protocol for the analysis of potential ecological interactions at the individual level from single DNA samples. Our results demonstrate that common DNA extraction methods can be successfully used to amplify a variety of taxonomic targets from a single individual. For simplification of approaches, ITS recovers a wider range of plant taxa than rbcL, and provides better taxonomic resolution, while simultaneously amplifying fungal taxa. However, there were taxa uniquely detected by rbcL and missed by ITS. We identified common potential interactions, *e.g.*, the frequent detection of *Wolbachia*, and interesting potential agents of biocontrol such as the detection of the fungus *Beauveria* in several samples of emerald ash borer. Data from pollinator larvae generated detections from a much wider variety of plants than either spongy moths or emerald ash borers, reflecting their role within ecological communities. The major challenge posed by this method is to establish whether detections correspond to true ecological interactions, or to co-occurring signals that may result from surface contamination of the specimens with DNA from co-occurring taxa that do not directly interact.

### Confirming target insect identification

DNA barcoding is an efficient and well-known method for the identification of insects (*Jung, Duwal & Lee, 2011*; *Park et al., 2011*; *Boehme, Amendt & Zehner, 2012*), and COI is widely used as the primary marker for taxonomic identification of animals (*Hebert et al., 2003*). Here we use it with a dual purpose: to confirm the identity of the host and to identify potential metazoan parasites. Unsurprisingly, host DNA dominated the sequencing datasets, with only trace amounts of DNA from other organisms, in terms of relative read abundances (RRA). We did use strict filtering protocols from negative controls which

might have resulted in artificially reduced detections of non-host taxa. The alternative, a more relaxed filtering, or not filtering but tracking potential contaminants, could raise issues around non-relevant detections (see below discussion of environmental DNA). Among emerald ash borer samples, we detected four orders of arthropods other than the host itself: two families of mites (Oppiidae, order Sarcoptiformes, and Eupodidae, order Trombidiformes), and one family of gall midges (Cecidomyiidae, order Diptera). Gall midges are primarily plant pests, and while some species can act as parasites of other arthropod larvae (*Hayon, Mendel & Dorchin, 2016*), we could not identify these further. These are likely co-occurring taxa rather than parasites. Notably, we detected a family of parasitoid wasps (Ichneumonidae, order Hymenoptera) in larva size 2, larva size 3, and pupa. Ichneumonids have previously been observed to parasitize emerald ash borers (*Duan et al., 2009*) making these highly likely cases of true parasitism.

The widest variety of taxa was found in the pollinator data where 24 occupants were detected, including aphids (probably kleptoparasites), and the wasp *Apocrita* sp., which may use the homes set out for non-colonial bees. The criteria used to determine what constitutes an "occupant" was arbitrary, and read abundances may vary depending on many factors other than true abundance and biomass (such as sequencing depth, primer-binding efficiency, *etc.*). However the genomic extracts used here had previously been screened and were selected based on having a clear "signal", thus we hoped to limit the challenge. However, we still encountered a small number where the ID of the occupant was unclear. In part this is due to a much larger reference library being available in this analysis than in a previous one. It may also be due to a common challenge where hymenopteran DNA is notoriously hard to sequence when in mixed templates. This is hypothesized to reflect the extremely high AT content of the mitochondrial genomes of Hymenoptera (*Clare et al., 2008*) which may lead to weaker PCR amplification in mixed templates where a more GC rich template is available. Among those where ID was clear, we did recover a reasonable set of taxonomic identifications, with common home occupants correctly labeled as "occupant", and known parasites as secondary signals. We suggest that these criteria be adapted to case-specific values for each new analysis as any such criteria will vary with sequence depth, primer binding *etc.*, and analysis for ecological objectives would be better aimed at screening unpooled larva, even though this increases analytical costs, to better control for occupancy by multiple species.

## The detection of potential insect-plant interactions

We used two markers to detect plant taxa: ITS and rbcL. ITS was amplified using two different primer pairs in our study. While the forward primer ITS3 was designed specifically for fungal taxa (*White et al., 1990*), the primer pair ITS-S2F/ITS4 is known to cross amplify fungi and plants, making it a useful general marker (*Chen et al., 2010*; *Cheng et al., 2016*). It may be most appropriate for use when plants are not the only target. For example, we used this primer set with spongy moths where there were a variety of host trees, and we were also interested in fungal amplification. In this case we did amplify a variety of plants and fungi, but the host tree was only identified once. While we were not specifically targeting plants, it was somewhat surprising that the host was not often

recovered, and may be linked to the strict filtering of our data, and the age and preservation of the specimens (extracted more than two years after collection and specimens not frozen immediately). In contrast to ITS, rbcL does not readily cross amplify most taxa outside of the plants, other than green algae or cyanobacteria (neither of which were expected in our samples). In spongy moths, rbcL identified 15 orders of plants compared to 13 with ITS, but only eight orders were common between the two. Unlike ITS, rbcL did successfully detect the host tree in 11 of the 18 samples. In pollinators the number of detections was much higher with ITS, recovering far more taxonomic diversity than rbcL. Individual pollinator larval samples contained DNA identified as up to 20 plant genera (rbcL) or more than 20 plant genera (ITS).

## The detection of potential insect-fungal interactions

The primer pair ITS3/ITS4 is less likely to amplify plant taxa and was used to amplify fungal DNA from emerald ash borers, where the tree host (ash) was known and, thus, devoting sequencing power to plant identification was not useful. Additionally, ITS3/ITS4 primers are known to have better specificity and amplify a greater variety of fungal taxa in comparison to other ITS primers (*Yu et al., 2022*). Most of the fungi detected within the emerald ash borer samples are associated with soil (*e.g.*, *Fusarium* sp.), decaying plant matter (*e.g.*, *Yamadazyma* sp.), lichens (*e.g.*, *Candelaria* sp.), or plant pathogens (*e.g.*, *Diaporthe* sp.), and not likely the larvae, pupae, or imagoes themselves, suggesting this is co-occurrence data rather than direct interactions, though it may be ingested material. One of the most commonly identified fungi is *Alternaria* sp., a genus of plant pathogens, involved in decomposition, but also ubiquitous in the environment and found almost everywhere (and considered a general environmental contaminant of airborne eDNA studies, *E. Clare personal observation*). However, one fungus of interest that was detected in two samples (one larva size 2 and one adult beetle) is *Beauveria* sp. (order Hypocreales). *Beauveria* spp. are entomopathogenic fungi that are known to successfully infect and neutralize emerald ash borers. Although *Beauveria* occurs naturally in the environment (*e.g.*, in soil; *Rehner et al., 2011*), biocontrol products containing *Beauveria* (*e.g.*, FraxiProtec) are also commercially available and have been shown to significantly reduce emerald ash borer populations (*Srei et al., 2020*). This detection here suggests some biocontrol may be ongoing and detectable by this analysis.

## The detection of potential insect-microbial interactions

Similar to the fungi detected by ITS, most of the bacteria associated with our samples are ubiquitous throughout the environment, and can be associated with soil (*e.g.*, *Massilia* sp. in spongy moth samples, and *Terriglobus* sp. in emerald ash borer samples), plants (*e.g.*, *Capsulimonas* sp.), lichen (*e.g.*, *Lichenihabitans* sp.), or the rhizosphere (*e.g.*, *Neorhizobium* sp. found in emerald ash borer samples). However, one notable bacterium found in all hosts is *Wolbachia* sp. (order Rickettsiales). *Wolbachia* sp. are intracellular parasites that can infect many insects and nematodes (*Tagami & Miura, 2004*), and are associated with reproductive abnormalities, such as the feminization of genetic males, parthenogenesis induction, and reproductive incompatibility (*i.e.*, through cytoplasmic incompatibility

between hosts infected by different strains or between uninfected/infected hosts; reviewed in *Werren, 1997*). However, it has recently been shown that *Wolbachia* might confer some protection to their host against viral infections, at least in dipterans (*Cogni et al., 2021*; *Pimentel et al., 2021*). *Wolbachia*'s ability to induce genetic incompatibility in populations coupled with the ability for hosts to transmit the infection vertically and horizontally means it may have potential biocontrol applications (*Werren, 1997*; *Zabalou et al., 2004*).

## Challenges of detecting parasites from hosts

We attempted to use both COI and 18S to detect parasitism. Parasite detection is extremely challenging for a number of reasons. First, differentiating true parasites from environmental DNA contamination of the body is nearly impossible. Second, without destruction of the sample, detecting internal parasites is rare (*e.g.*, sometimes detected from faeces or during voucher saving extraction methods) and most signals will likely be from surface contamination. A challenge with COI is the swamping of secondary signals with host DNA which is both more abundant and co-amplified, and most primers in DNA barcoding have been designed to minimize taxonomic bias. An alternative target is 18S where some primers show higher affinity for nematodes, intracellular parasites, *etc.* However, short 18S regions suitable for most high throughput sequencing on platforms such as Illumina provide limited taxonomic resolution. To address this gap, we sequenced long 18S amplicons on the MinION instrument (Oxford Nanopore, Oxford, UK) which provides much lower sequencing depth but can accommodate long read lengths at low cost, even with a high negative detection rate.

We tested this approach on emerald ash borer specimens and, interestingly, the 18S region detected the greatest diversity of interaction types (Fig. 2), including animals (nematodes and arthropods, which were expected), but also some fungi, and a parasitic alveolate. All fungal orders detected using 18S were also detected using ITS, but, despite being a longer amplicon length, the taxonomic resolution offered by 18S was the same or worse than ITS, likely due to a combination of generalist primers lacking resolution and incomplete reference libraries (*e.g.*, taxa recovered by both markers but assigned with ITS to genera such as *Diaporthe* sp., *Niesslia* sp., and *Cryptococcus* sp., were only resolved at a higher taxonomic level with 18S). This is consistent with other studies that have found the ITS marker to provide a more extensive view of fungal diversity and allow more precise identifications to be made compared to the 18S marker (*Lord et al., 2002*; *Liu et al., 2015*). One notable detection was the family Cordycipitaceae (order Hypocreales), which contains the previously noted fungal biocontrol genus *Beauveria*, independently confirming its presence in the same samples, and providing support to its detection when we treated it with caution in ITS data. While the recovery was much lower in taxonomic richness, the lower resolution of the 18S region might be a useful screening tool for other interactions.

The non-host arthropods detected belong to order Hymenoptera. Potential interactions with nematodes (order Rhabditida) detected include *Rhabditolaimus* sp. (bacteriophagous and commensal with other beetles, *e.g.*, *Scolytus multistriatus*; *Ryss & Polyanina, 2022*), *Panagrellus* sp. (free-living nematodes associated with many habitats, including insect frass; *Ferris, 2009*; *Srinivasan et al., 2013*) and Neotylenchidae (members can be parasitic,

*e.g.*, *Deladenus proximus*; *Zieman et al., 2015*). Finally, the parasitic alveolate detected (*Cryptosporidium* sp.) has been reported to infect humans and animals including insects (*Helmy & Hafez, 2022*).

## Novel challenges in detecting species interactions

Environmental DNA (eDNA) refers to DNA shed by organisms (*e.g.*, through cells, excreta, *etc.*) that is deposited throughout the environment, including in soil (*Marquina et al., 2019*; *Foucher et al., 2020*), water (*Uchida et al., 2020*), and air (*Clare et al., 2021*; *Clare et al., 2022*). eDNA can also be spread between organisms directly through contact. For example, insects leave eDNA behind on the surface of plants they inhabit/consume (*Kudoh, Minamoto & Yamamoto, 2020*; *Allen et al., 2023*) and this can be picked up by another insect without any actual interaction. This was demonstrated by *Huszarik et al. (2023)*, who found insect eDNA was spread to spiders who occupied the same wet pitfall trap as an exotic insect that was not consumed by the spiders. It is thus inevitable for target organisms to become coated in eDNA simply by occupying their natural environment or during collection/storage depending on the medium (*e.g.*, pitfall traps, malaise traps, tubes, ethanol, *etc.*; *Shokralla, Singer & Hajibabaei, 2010*; *Huszarik et al., 2023*). While eDNA's ubiquitous nature allows for its use in biodiversity and community composition assessments (*Clare et al., 2022*; *Macher et al., 2023*), it can complicate analyses of interactions through metabarcoding. More specifically, it is not clear how long eDNA remains detectable and it is unclear whether eDNA contamination on the surface of insects or other target organisms poses a challenge to the assessment of species interactions. Surface contamination could make it difficult to determine true interactions from mere co-occurrence, but only if eDNA regularly persists, something which has not been robustly tested.

Some of the detections recorded here were unexpected. For example, when attempting to confirm target hosts, we often detected other insects which are not likely directly associated with the target. In spongy moth samples we detected six insect orders, including other plant pests (*e.g.*, *Epuraea* sp. and *Pineus* sp.) and predatory insects (*e.g.*, *Hemerobius* sp. and *Lestodiplosis* sp.). Spongy moths may encounter other plant pests (either directly or indirectly) on the host trees and may have even been predated on by some of the insects detected like lacewings (*Hemerobius* sp., order Neuroptera). For example, green lacewings are generalist predators that have been observed to feed on lepidopteran eggs and small larvae (*Brown & Cameron, 1982*; *Tauber, Tauber & Albuquerque, 2009*). However, it is impossible to confirm whether these are predatory interactions or simply encounters with other insects in the environment. Interestingly, in one spongy moth sample from a pine tree a pine-feeding aphid (*Pineus* sp.) and its predator (*Lestodiplosis* sp.) were also detected (*Gagn & Havill, 2020*). While it is unlikely this was a direct interaction with the spongy moth, it suggests the detection of another local interaction occurring on the host tree, a fascinating ecological process to be recorded in surface contamination by environmental DNA of nearby organisms, though unrelated to the objective of the analysis. Despite the potential accumulation of DNA signatures with time from environmental sources, the actual profile of co-occurring DNA signatures was remarkably stable with life stage
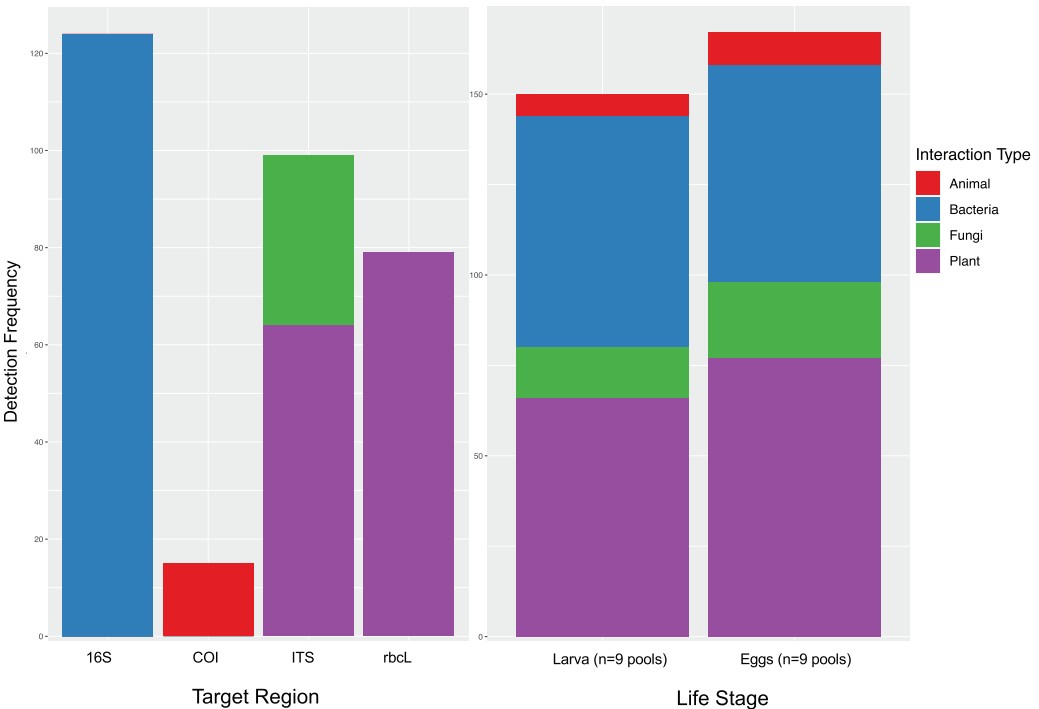

**Figure 4 Highly similar co-amplification in eggs and larval spongy moths.** The recovery of various targets by target amplified region (left) shows high variability in taxonomic richness. Interestingly the profile of taxonomic recovery of co-amplified taxa is highly similar between egg masses and larval stage spongy moths (right) suggesting little change in the richness of interacting taxa over development.

(*e.g.*, see Fig. 4 for spongy moths) which suggests this surface contamination risk may be minimal.

The effect of potential eDNA contamination was larger in plant identifications, and this may reflect the more robust cells and pollen of plants. We commonly identified DNA from trees and plants that were not noted during spongy moth collection with both ITS (*e.g.*, *Salix* sp., *Poa* sp., *etc.*) and rbcL (*e.g.*, *Rosa* sp., *Picea* sp., *etc.*). These are possibly a result of pollen from nearby plants sticking to the fuzzy surface of egg masses or to the hairs/surface of the larvae and, in the case of the larvae specifically, we may also be detecting trees that previously hosted the larvae before collection. Among bees it may be impossible to differentiate direct provision of the larva by the mother, material used in nest construction, and casual environmental contact. When the concept is expanded to colonial honeybees, the honey itself is a well-known reservoir of plant and microbial DNA (*Hawkins et al., 2015*; *Balzan et al., 2020*; *Ribani et al., 2020*; *Pathiraja et al., 2023*).

Most of the fungi detected are ubiquitous throughout the environment and are often found in soil (*e.g.*, *Sporobolomyces* sp.) or plants (*e.g.*, plant pathogens like *Alternaria* sp.). However, other fungi (*e.g.*, *Punctelia* sp.) and algae (*e.g.*, *Coccomyxa* sp., *Trebouxia* sp., and *Symbiochloris* sp.) that often form lichen complexes (*Sanders & Masumoto, 2021*) were also identified as part of the spongy moth samples by ITS, rbcL, or both. Thus, many

identifications made from whole bodies are not target interactions but seem to reflect surface level contamination from co-occurring species through environmental DNA.

There are several potential solutions to this problem of surface contamination. First, it may be prudent to wash specimens prior to DNA extraction (*i.e.*, in a bleach solution) to remove surface DNA (*Binetruy et al., 2019*; *Hausmann et al., 2021*; *Huszarik et al., 2023*), particularly if they have been stored in a common location (*e.g.*, a malaise trap collection pot). This was done to the Hymenoptera larva prior to their original extraction and is thought to remove surface DNA but may limit the detection of fungi and microbes associated with that ecology, though we noted non-target identifications there as well. Similarly, the gut of an insect might be dissected out to determine ingested plants and gut microbiota and parasitism, which are more likely to reflect direct interactions than the causal contact detected by surface environmental DNA. This would also likely decrease the false negative rate in the detection of gut parasites like nematodes by better exposing them to DNA extraction and reducing the proportion of host DNA in the mix. This, however, would cause substantial or complete destruction of the specimen. In marine systems, where diet analysis must consider actual ingested material *vs.* DNA in swallowed water, one suggested solution has been to analyze DNA in water samples along with DNA from stomach contents to evaluate the potential errors (William. O.C. Symondson, 2012, personal communications). Some version of this, considering body surface and gut separately, bee home tube and larva, or host tree tissue separately from individual pest might similarly help establish likely error rates, though this would also increase the false negative rate: if individual A interacts with parasite B, both may leave DNA in the environment. Thus, removing signals that are present in the environment may be overly conservative. Similarly, restricting our detections only to those for which an ecology has already been established limits the discovery of new true interactions. It may instead be argued that the definitions of symbiome *vs.* holobiont, interactions *vs.* co-occurrence are too strict a categorization of relationships which likely fall along a continuum of effects. Strongly interacting species, those with clear coevolutionary "Red Queen" relationships fall at one end, and neutral co-occurring species at the other, but between them are a range of weak interactions, secondary interactions, and tertiary interactions, which form the more complex roles within ecosystems.

## Pooling of individual PCRs

One consideration in multi-gene analyses is the cost associated with tracking individual samples with unique tags during sequencing (frequently called indexing). Pooling target regions before indexing samples significantly reduces the costs of analysis, but there are challenges in PCR 2 with the addition of such indexes (*e.g.*, shorter amplicons can become overrepresented during PCR 2). We tested this hypothesis by comparing pooled and unpooled samples from our most taxonomically complex PCRs generated from larval pollinators. Our data suggest that pooling greatly reduced taxonomic representation with many of the most common taxa in unpooled samples completely missing in pooled ones. We observed the same effect in all gene regions suggesting it was not confined to underrepresentation of specific amplicons. The most likely explanation is that pooled

libraries lacked sequencing depth compared to individual-PCR libraries: while the pooled libraries contained four times more information (the four markers used for each sample), they were added equimolarly to the sequencing mix, meaning that each pooled library was allocated only one fourth of the sequencing power (which could translate to lower "molecular sampling effort") compared to individual-PCR counterparts. A more efficient approach may be to amplify samples using PCR 1 primers with unique codes or tags added at the 5′ or 3′ end so that they can be indexed with the same tag during PCR 2 and pooled after (*Binladen et al., 2007*). This creates a bioinformatic challenge but would reduce indexing steps, albeit increasing laboratory costs (many versions of the same primer, one with each tag). Alternatively, one-step PCR approaches where fusion primers (which include the primer sequence, a sequencing adapter, and nucleotide tags) are used can also eliminate the need for indexing because of the presence of nucleotide tags at the cost of reduced PCR efficiency and increased primer cost (*Bohmann et al., 2022*). Multiplexing approaches where multiple primers are used to amplify different target regions simultaneously can also be employed to increase efficiency (*Zhang et al., 2018*). However, difficulties may arise in multiplexed PCRs, such as the formation of primer-dimers and the over-representation of some amplicons over others (*Khodakov, Wang & Zhang, 2016*), particularly where different extension times are required.

## CONCLUSIONS

In this analysis we compared the recovery of a variety of markers for the description of individual ecological interactions for common insects and to evaluate the performance of different target regions for the identification of potential ecological interactions in these taxonomic groups. Our data suggests remarkable similarity in identified taxa between life stages (*e.g.*, eggs, larvae, pupae, and adults) for spongy moths but shows some difference between life stages in the emerald ash borers. We recovered a very large number of potential floral interactions in native and non-native resident pollinators which may suggest interesting ecology in the provision of plant resources by adults to larva. Our data also highlights the potential environmental contamination of specimens in ecological analyses generating false positive "interactions" from contact with environmental DNA. This requires creative solutions to estimate and mitigate the challenges depending on the analysis of interest.

## ACKNOWLEDGEMENTS

We are grateful to the collection team of the Canadian Center for Biodiversity Genomics who provided access to all specimens and Sage Handler who provided DNA extracts used in this analysis. In particular, Jayme Sones and Sean Prosser who contributed materials and Paul Hebert, who was the PI on the grants, helped to obtain access to the specimens and resources.

### Funding

This work was supported by The Natural Sciences and Engineering Research Council of Canada through the Discovery Grants Program, The Government of Canada's New Frontiers in Research Fund (NFRFT-2020-0073), and through Genome Canada and Ontario Genomics (OGI-208). The funders had no role in study design, data collection and analysis, decision to publish, or preparation of the manuscript.

### Grant Disclosures

The following grant information was disclosed by the authors:
The Natural Sciences and Engineering Research Council of Canada.
Discovery Grants Program, The Government of Canada's New Frontiers in Research Fund: NFRFT-2020-0073.
Genome Canada and Ontario Genomics: OGI-208.

### Competing Interests

Dirk Steinke is an Academic Editor for PeerJ. Daniel Marquina is employed by AllGenetics & Biology SL.

### Author Contributions

- Nicole D. Borsato conceived and designed the experiments, performed the experiments, analyzed the data, prepared figures and/or tables, authored or reviewed drafts of the article, and approved the final draft.
- Katherine Lunn conceived and designed the experiments, performed the experiments, analyzed the data, prepared figures and/or tables, authored or reviewed drafts of the article, and approved the final draft.
- Nina R. Garrett analyzed the data, authored or reviewed drafts of the article, and approved the final draft.
- Alejandro José Biganzoli-Rangel analyzed the data, authored or reviewed drafts of the article, and approved the final draft.
- Daniel Marquina performed the experiments, authored or reviewed drafts of the article, and approved the final draft.
- Dirk Steinke conceived and designed the experiments, authored or reviewed drafts of the article, and approved the final draft.
- Robin Floyd performed the experiments, analyzed the data, authored or reviewed drafts of the article, and approved the final draft.
- Elizabeth L. Clare conceived and designed the experiments, authored or reviewed drafts of the article, and approved the final draft.

### DNA Deposition

The following information was supplied regarding the deposition of DNA sequences:
The sequences are available at GenBank in the SRA BioProject: PRJNA1150198.

## Data Availability

This is a methods article.

## Supplemental Information

Supplemental information for this article can be found online at http://dx.doi.org/10.7717/peerj.18906#supplemental-information.

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
