# Peer review of "Identification of potential insect ecological interactions using a metabarcoding approach"

_PeerJ, doi:10.7717/peerj.18906_

## Round 0.1 · original submission · Major Revisions

· Academic Editor

Major Revisions

This manuscript is on a subject of interest to a borad community and shows great potential, however it lacks cohesiveness and clear links between stated aims, results, and conclusions. Both reviewers have made extensive suggestions for improvement. Please revise the manuscript in line with these comments and provide a detailed response for each of their suggestions.

Reviewer 1 ·

Basic reporting

Figures: The x axis of life stages is hard to read in Figure 2 and the general font and picture resolution of all the figures could be increased.

Line 90: Add the word been. “it may have been present…”

Experimental design

Line 51: Within this section or possibly in its own section, it would be useful to touch on the idea described below how eDNA is not necessarily indicative of an interaction per say (airborne eDNA deposited, previous meal of collected species, etc.). This way the concept is introduced early.
Line 84: What is an example of an ecological interaction that could play a role and be shown molecularly.
Line 111: Please match the other sections and include information about how molecular symbiont work is useful for this species.
Line 136: The bulk of the introduction does not feel like it built toward these objectives. I think more attention should be paid to the benefits gained from exploring these objectives and possibly less information on the target species (could be included in methods instead of intro).
Line 136: More molecular information could be added to the introduction as well. As of now it is fairly vague what will be accomplished and what DNA is actually being explored from these species.
Line 156: How were these samples collected (from all organisms not just the moths)? Were sterile gloves used? If not, it is possible some of the DNA on these species could be a contaminant.
Line 160: The moths were pooled but the emerald ash borers were not, correct? In this case, a sample is a single beetle? This information is included on line 183, but to match the moth section, I would recommend adding in what a sample is in the other sections.
Line 167: Were these parasites identified prior to extraction?
Line 182: Out of curiosity, why were 4-5 moth larvae used but only parts of the emerald ash borer? I assume the sizes varied?
Line 189: Could you add more detail as to what makes up these 100 PCRs? Starting with 18 samples in total and then amplifying rbcL, CO1, IT2, and 16S, I am not sure how 100 was achieved based on Table 1. Or does this also include the pooled numbers. In general, a figure showing the full sample reactions may be useful.
Line 197: What is the concentration of initial primer or final primer for all assays? Could just add to table 1 as well.
Line 201: For my own curiosity, why the substantially fewer cycles for CO1?
Line 281: Could you please include more detail here? The pooled PCR products were all combined into a single sample? Would this mean the moths had a pooled sample that was approximately 200ul of sample in a tube? How much of this 200ul was then used for analysis?
Line 314: Was the same % identity used here as above?

Validity of the findings

Line 339: Were there any samples where multiple ASVs were above 100 reads?
Line 340: Could you please explain the no occupancy sample further? This is saying that there is no way to say what the host species is based on the sequencing results correct (or am I misunderstanding)? Does not having a distinct source of DNA based on tissue samples cast doubt on the approach as a whole or is it due primarily to reference limitations?
Line 341: I will discuss this further below, but I think it is not possible to fully say a detection of DNA is an interaction. I believe anything listing an interaction should say “possible interaction”.
Line 360: Would there be any benefit to an OTU clustering approach to account for the lack of species specific identifications? Do the authors expect multiple species within a genera that this would help elucidate? Were the lack of identifications due to reference library limitations?
Line 437: How many unique taxa were ID’d just in the pooled sample?
Line 450: I am curious if only IDing to genera primarily limits the comparison, would there be a way to check how each of these primers preforms going down to a species level? Also, what is the mechanism behind these differences, performance of the assay or reference library availability. It would be helpful to explore how reference library information changes between the markers used.
Line 450: In general, more discussion on differing primer performance would be useful. Could a statistical approach such a permanova based on dissimilarity indices be used to demonstrate raw biodiversity differences between assays?
Line 454: I think one of my strongest concerns is brought up by this point and discussed below. There is a bit of a disconnect between the large number of caveats on eDNA interactions and how the interactions are discussed throughout. In general this is no real way of knowing if an interaction actually took place. As such, the language used to discuss these interactions needs to be softened throughout. It should refer to potential interactions. eDNA is great for large scale sampling to narrow down for conventional approaches and interactions such as these are great examples. We can say a potential interaction occurred, but more detail needs to be collected to confirm.
Line 489: Example where it should be potential insect-plant interactions.
Line 603: All great points in this section, but that uncertainty is not shown when discussing the results.
Line 633: Do the plant genera and sampling time align with any pollination seasons or specific syndromes? We know plant airborne eDNA matches plant phenology strongly.
Line 675: Significantly implies statistical analysis.
Line 694: Would increasing the read count via a NovaSeq also be an alternative?
Line 705: The conclusion is at odds with the introduction and title. With “the extreme level of environmental contamination” being one of the key take-homes while the title and introduction only refer to interactions that could be gained. It may be beneficial to reframe the introduction around the challenges of detecting interactions.

Additional comments

General Comments

This manuscript explores the comparison of several DNA markers for assessing the amplification of co-occurring eDNA on several economically important insect species. In general, the manuscript reads well, the methods are well designed, and the results are compelling. My main comments heavily focus the interpretation of these results. First, the authors do a good job listing the caveats to using eDNA for interactions, but the language describing the potential interactions needs to be softened and made consistent throughout. Second, reformatting the introduction to discuss these potential issues would better suite the majority of the discussion and results in the rest of the manuscript better. So much text focuses on symbiont community identification when the final results end up having a lot of uncertainty. Lastly, I would like to see additional analysis and details for comparing the different primer sets.

Reviewer 2 ·

Basic reporting

Abstract – “challenging to morphology” doesn’t make sense, perhaps “are morphologically challenging to identify” or “challenging for morphological identification” or similar.
Abstract – “The challenge has shifted…” the last part of this sentence should be re-phrased.
Abstract – suggest a comma after Agrilus planipennis),
Abstract – larva should be larvae here I think.
Abstract – what parasitic phyla? This reads such that these parasitic phyla aren’t plants, fungi, microbes or insects, is that correct?
L20 - perhaps add something here about why identify these interactions is important.
L25 – is “being eaten by it” really included in the definition of symbiome?
L45 – Suggest to keep language consistent throughout, earlier in this section a sample is a DNA extract and then later (e.g. L47) is back to “samples”.
L48 – “characterize the biological interactions” – I’m not sure this is accurate, see comments throughout regarding interactions vs. environmental DNA (eDNA).
L54 – this really needs some referenced examples of where genes have been used specifically to demonstrate interactions, rather than for say community diversity assessments.
L122 – if collected as larva how do you have a sample that identifies the larva and provides ecological interactions of a pollinator when the sampled individuals have not been functional pollinators yet?
L135 – as with the abstract, clearly state the application here, “a general metabarcoding approach which can be applied to a variety of insect targets” – so you are just trying to identify the insects with this method? Or insect-insect interactions only?
L144/Methods Generally – It’s unclear if these are specimens from within a collection or DNA extracts from within a collection? If whole specimens, how many were pooled together, from what locations etc. Are the samples from L153 onwards the samples from these collections?
L182 – how would a single egg be considered to represent an individual? Do these moths only lay a single egg?
L186 – suggest ending this section with a reminder of total number of DNA extractions to be used going forward per species group for clarity.
L189, L247 – so 100 PCRs per every extract? 294 PCRs per extract? The repeated stating of total PCRs here doesn’t help the reader, are these all that was sequenced? How many per target? How many per sample? How many replicates per target per sample?
L192 – reference required for CS1/CS2 primer modification.
L195, L203, L209, L224, L232 – references aren’t given for these primers (but are in the table), but then the references in the text appear to be for the reaction conditions and not necessarily the primers. E.g. L203, those primers aren’t designed in Cheng et al. 2016, with one from Chen et al. 2010 (for plants) and ITS4 from White et al. 1990 for fungi.
L330 – OUT instead of OTU
L472 – “Gall midges are...” this sentence needs a reference(s).
L496 – this sentence doesn’t make sense, “In this case….”.
L525 – how do you know this is naturally occurring and not co-occurring via eDNA, use of such products in neighbouring areas etc.?
L549 – how is this statement not true of every other taxa detected herein?
L621 – this could be to do with the surfaces of eggs vs larva (for example) and/or their locations as to how much eDNA is retained/accumulated.
Figure 1 – in the caption “suggesting a highly similar core microbiome across life stages”, this is misleading. It is not known that this is the hosts microbiome vs. eDNA/co-occurrence, as per the discussion.
Figure 3 – in the caption “ITS primers targeting primarily plant DNA” are those primers really more primarily targeted to plant DNA?
A reminder to please always introduce acronyms on first use, please correct throughout (e.g. HTS, QC, ASV, GBIF, OTU etc.).
How were the figures generated?

Experimental design

How well can we expect samples from existing collections are going to represent ecological interactions other than internally though digests etc.?
How are pollinator interactions to be studied from larva collected from bee homes? Is this an appropriate sample choice to investigate a pollinator's ecological interactions? Sample type wasn’t detailed in the introduction. Any eDNA on the exterior would be contributed through wind, surface contact, contact with others etc., how is detection of these species useful to know in context of investigating the interactions of a pollinator such as plants visited, symbiotic bacteria of the adult etc.? Were environmental control samples collected for comparison to the insects sampled (e.g. swab inside the home, exterior of the home)? This would be useful to help differentiate possible true interactions and passive/eDNA co-occurrence taxa.
L186 – were extraction blanks also available to act as a control for the archived native pollinator samples?
ITS plant and fungal amplification – how confident are the authors that this primer pair works equally as well in plants as in fungi? This seems an unusual approach where typically a primer pair would be used in each group that is most appropriate for the broad amplification of taxa within those (very large) groups.
L224 – why here are the ITS plant and fungal primers (single primer pair) different to those used previously? L219 indicates that primer choices were changed but how the methods section is written this makes it difficult to follow, suggest a methods section re-structure.
L203/L224 - Spongy moths have ITS-S2F/ITS4 and emerald ash borers have ITS3/ITS4, both used for plant and fungal amplification, but ITS3 is in the discussion as specific to fungal taxa.
L275 – This is unclear, how many PCRs per how many samples and for how many targets were pooled? There needs to be more detail here since this is a core objective of the study and it is hard to know what was tested here.
L285 – all the reagents and equipment here have to be properly referenced (e.g. Ampure beads, qubit etc.).
L358 – were these counts of identified taxa after accounting for negative control read counts etc.? How likely is it that the algae for example are accurate detections and not false-positive/contaminants etc?
L466 – Primer choice also is likely to have biased the recovery of sequence data.
L510 – comment that the host tree was known so sequencing power expended on plant identification wasn’t helpful, was the point not to map and detect ecological interactions/detect various taxa? Surely this species is expected to have more than a single plant interaction?
L553 – this may have been a primer pair choice issue too if these primers aren’t very generic - how likely were parasites of the native pollinators to show up using COI when the primer pair designed for the hosts was used?

Validity of the findings

Development of a standardized protocol for the analysis of ecological interactions at the individual level from single DNA extracts is described as the objective of this work, but this reviewer cannot really see a standardised method within this work. Different DNA targets are used for different sample types, and additionally occasionally different primers for the same DNA target are used for different sample groups. Two sequencing platforms are used and different raw sequence data processing pipelines are deployed per DNA target. It may also be that different methods to identify taxa are used (ASV vs OTU, although this is unclear). Pooled versus individual sample libraries across the DNA targets is discussed but isn’t clearly described. Additional variation in metod includes not targeting plants in one species where they were specifically targeted elsewhere, occasionally a primer pair that amplifies both plants and fungi is used, and then elsewhere a primer pair more specialised for fungal identification only is used, and 18S is tested in one species group only.

Ecological interactions – how are these taxa detections, some of which are detected in a single instance in one sample per sample type, confidently reported as interactions? What is the context of an interaction here? Is an interaction simply contact with environmental DNA (i.e. could be passive)? And if so, if larva and eggs are co-situated is this still considered a genuine egg life stage interaction with plant DNA (for example) if the plant DNA on the egg is there through contact? Interpreting these taxa detections in the context of ecological interactions may be beyond my expertise here but some structure around what the authors determines an ecological interaction within this work should be described, particularly for sedentary samples like eggs. This is mentioned in the discussion, as to the challenge of determining detections as being ecological interactions or co-occurring taxa, but throughout the introduction, methods and results sections these detections are introduced, often described and/or reported, as interactions implying those taxa have been defined as representing ecological interactions. The sentence “Native bees interacted with a much wider variety of plants than either spongy moths or emerald ash borers, reflecting their role within ecological communities.” is clearly classifying all of the taxa detections as interactions across those insect groups but what is the basis for this (particularly given only larvae were used for bees)? Figure 4 caption is the “Highly similar interactions…”, how are these taxa detections known to be from interactions? Additionally “suggesting little change in the richness of interacting taxa over development”, how is this attributed to lifestage when it could simply be both life stages studied exist in the same environment and so the same taxa are detected from each? Do the eggs and larva exist in the same environment for this group? Assessing the feasibility of detection of a plant species know to exist in a sample area (for example) is one thing but that doesn’t confirm the nature of how that plant’s DNA came to be in a sample: ecological interaction or co-occurring taxa/eDNA

I am not sure the conclusions as written reflect the data as generated. The comment that similarity was identified between life stages in spongy moths but differed in emerald ash borers could well be influenced by the fact that varied DNA targets were deployed in each group, and that one group also included adults. The comment that pollinators interact with higher plant diversity is also misleading as for one of the groups plants were specifically not targeted for data generation. The data for comparison are not generated in the same way.

Additional comments

The authors have presented a large metabarcoding study targeting three insect groups of importance. From the abstract and introduction this work set out to evaluate different gene regions for ecological interactions, assess analyses at different life stages and compare pooling vs individual PCR methods for marker outputs. I believe there is a disconnect here, where the dicussion and results in context does not really align with the introduction and the objectives. The authors have put together a good sized dataset of samples and have a wealth of sequence data so my recommendation is re-working the manuscript to report this work in a more cohesive way.

Pest species that cause a lot of damage, but what is the importance or value of investigating their ecological interactions here? Use as a screening tool to predict outbreaks? But if you can predict an outbreak how does that help? L86 – “present mechanisms for population control.” – how?
Other than being easier to collect, what is the rationale behind studying larvae for ecological interactions, particularly for pollinator species and pest species most harmful in adult stage? L250 “Because we expect a larger range of plant interactions” – why? These samples are from larva that haven’t left the bee home, why would we expect a larger number of plant interactions for a pollinator in larval stage that hasn’t been flying around visiting various different species of plant? What does “actual occurrences” refer to here? Later “actual detections” is also used.

The current layout of the Method sections makes the work completed quite difficult to follow, particularly where the same target is described but different primer pairs are used. This is true also of the sequence data processing section where target data is processed using one of several different pipelines. It is understandable that this is the authors likely using the best pipeline per target but it makes this section a bit difficult to follow. Additionally clarification on whether the COI was OTU or ASV should be made (there are OTU thresholds set, but then these data are referred to later in the same paragraph as ASVs).
Many of the taxa detected in different samples are only seen once per sample type, how conclusive can these rare data be treated?
The individual vs pooled samples isn’t overly clear throughout. The methods section is very brief and reads as all samples have individual and pooled sample libraries for comparison. Then for the results section the taxonomic coverage for pooled PCRs was lower but this would be expected given the nature of pooling used and the likely impact on sequencing depth per target per sample. In the results section it is still unclear exactly which targets were pooled and for which samples (although presumably not 18S given the platform difference and rbcL also doesn’t seem to be included), or what the sequence counts and raw performance looked like per sample per individual vs pooled sample library etc. This is one of the main objectives of this work (as per the Introduction) and yet it isn’t well described or the results discussed in much detail.

---

## Round 0.2 · accepted · Accept

· Academic Editor

Accept

All of the reviewers' suggestions have been adequately addressed and the current version of the manuscript is ready for publication.